# Federated Fine-tuning of Large Language Models under Heterogeneous Tasks and Client Resources

**Jiamu Bai**[*]
Pennsylvania State University
jvb6867@psu.edu

**Daoyuan Chen**[*]
Alibaba Group
daoyuanchen.cdy@alibaba-inc.com

**Bingchen Qian**
Alibaba Group
qianbingchen.qbc@alibaba-inc.com

**Liuyi Yao**
Alibaba Group
yly287738@alibaba-inc.com

**Yaliang Li**
Alibaba Group
yaliang.li@alibaba-inc.com

## Abstract

Federated Learning (FL) has recently been applied to the parameter-efficient fine-tuning of Large Language Models (LLMs). While promising, it raises significant challenges due to the heterogeneous resources and data distributions of clients. This study introduces FlexLoRA, a simple yet effective aggregation scheme for LLM fine-tuning, which mitigates the "bucket effect" in traditional FL that restricts the potential of clients with ample resources by tying them to the capabilities of the least-resourced participants. FlexLoRA allows for dynamic adjustment of local LoRA ranks, fostering the development of a global model imbued with broader, less task-specific knowledge. By synthesizing a full-size LoRA weight from individual client contributions and employing Singular Value Decomposition (SVD) for weight redistribution, FlexLoRA fully leverages heterogeneous client resources. Involving thousands of clients performing heterogeneous NLP tasks and client resources, our experiments validate the efficacy of FlexLoRA, with the federated global model achieving consistently better improvement over SOTA FL methods in downstream NLP task performance across various heterogeneous distributions. FlexLoRA's practicality is further underscored by our theoretical analysis and its seamless integration with existing LoRA-based FL methods, offering a path toward cross-device, privacy-preserving federated tuning for LLMs.

## 1 Introduction

Large Language Models (LLMs) have propelled advancements in natural language processing (NLP), offering breakthroughs in various tasks [45]. Finetuning LLMs on specific datasets enhances their applicability [12, 33], yet collecting such datasets raises concerns regarding cost and privacy [29, 6].

Researchers have turned to Federated Learning (FL) as a means to fine-tune LLMs using more data across distributed clients without compromising data privacy [28, 2, 43, 32]. In these settings, parameter-efficient fine-tuning techniques [34], particularly Low-Rank Adaptation (LoRA) [16], become attractive for reducing computational and communicational burdens [44, 41, 8].

---

[*]Equal contribution. Work done during Jiamu Bai's internship at Alibaba Group.

38th Conference on Neural Information Processing Systems (NeurIPS 2024).

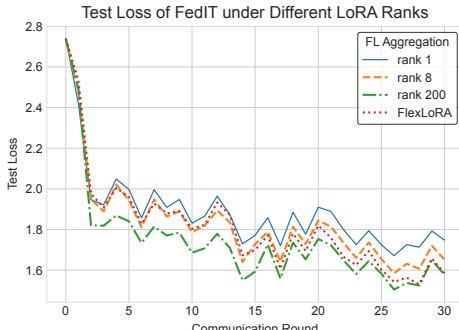

Figure 1: Test loss of FlexLoRA and FedIT [43] across communication rounds under LoRA ranks of 1, 8, and 200. FlexLoRA demonstrates adaptability in an "extreme heavy tail" scenario and increasingly aligns with the performance of FedIT at the highest LoRA rank as rounds progress. Implementation details are in Appendix A.

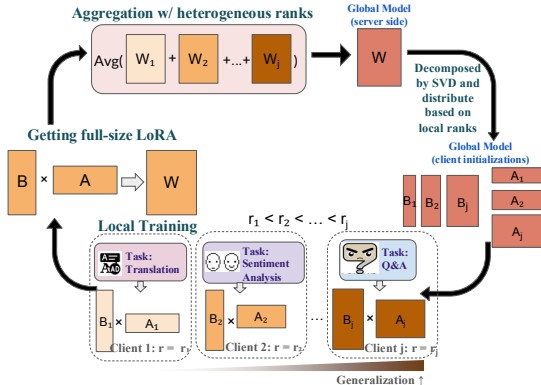

Figure 2: Illustration of FlexLoRA. The server initially constructs a full-size LoRA weight, which is then averaged across client-contributed weights with different ranks. The aggregated global weights are decoupled via SVD and sent back to clients.

Despite its efficiency, LoRA's use in FL is challenged by the heterogeneity of downstream tasks and available resources among clients, especially in cross-device scenarios [39, 5]. Traditional FL methods often suffer from "bucket effect", converging to the use of the smallest viable LoRA rank for all clients, even though many clients typically have more resources that remain underutilized. A small LoRA rank, optimizing weights in a task-specific manner [16], can be sensitive to heterogeneous data distributions and compromised generalization when applied to all clients, as evidenced in Figure 1. Ideally, we hope all clients can fully leverage their advantages by sizing their local LoRA ranks with their resources to contribute models with less task-specific but more generalized knowledge.

To address these challenges, we propose FlexLoRA, a simple yet effective FL aggregation scheme that enables the mixture of diverse LoRA weights across individual clients. It accounts for local resource and task differences and aims for a well-generalized global model. With the heterogeneous aggregation and redistribution of weights through Singular Value Decomposition (SVD), FlexLoRA ensures all clients contribute effectively, regardless of resource capacity. Thanks to the simplicity, FlexLoRA can be pluggable into a series of LoRA-based FL methods, unlocking their potential to leverage available yet under-utilized resources to contribute more generalized knowledge via larger LoRA ranks, which is also supported by our theoretical analysis.

Our empirical study, simulating a cross-device federate fine-tuning scenario with thousands of clients on various of NLP tasks [39] and resource distributions, underscores the real-world applicability of FlexLoRA. Notably, FlexLoRA can be readily applied to several SOTA FL baselines in a plug-and-play manner and achieves significant performance enhancements, including 3.1% and 4% improvements in zero-shot Rouge-L scores and language understanding tasks such as overlap extraction and textual entailment, demonstrating robust generalization capability. We further conduct an extensive study on the aggregation scheme and scalability of FlexLoRA, furnishing a more nuanced understanding of the underlying mechanisms that facilitate its effectiveness

Our contributions can be summarized as follows:

- We propose a simple-yet-effective scalable method to fully leverage local client resources for enhancing the global model's generalization ability, supported by both theoretical analysis and extensive empirical evidence.

- To our knowledge, this is the first work to demonstrate the feasibility of federated tuning of billion-sized LLMs across thousands of NLP tasks in large-scale, resource-heterogeneous scenarios.

- We explore the interplay between LoRA ranks, client numbers, specific heterogeneous language tasks, and resource distributions, offering practical insights. Our code is made available at *https://github.com/alibaba/FederatedScope/tree/FlexLoRA*, inviting further research and application in real-world cross-device FL for LLMs.

## 2    Related Work

**Parameter-Efficient Fine-tuning of LLMs.** The computation and storage demands of traditional fine-tuning processes have spurred the development of parameter-efficient fine-tuning (PEFT) techniques such as adapter and prefix tuning [15, 23]. Among existing PEFT techniques, we choose to employ LoRA due to its simplicity and outstanding performance [24, 18]. Despite this, our aggregation scheme can be easily extended into other PEFT methods by replacing the LoRA weights with their alternative weights to be tuned.

**PEFT in Federated Learning.** PEFT techniques have been integrated into FL to minimize communication costs and maximize efficiency. Several works employ LoRA for local model updates within an FL framework [2, 43, 44, 41, 19, 8, 26, 30, 38, 42]. For instance, [43] combines LoRA-based local updates with FedAvg for model aggregation, while [2] intersperses sparse finetuning with LoRA fine-tuning for improved initialization for LoRA in FedAvg. [36] proposes a technique to improve LoRA performance in FL scheme, [42] exploits performing SVD on pretrained model weights to resolve data heterogeneity, and [32] reduces communication cost through zeroth-order optimization. Distinct from these methods, our work, by introducing a simple yet effective aggregation scheme, leverages heterogeneous client resources to enhance the generalization and natural language understanding of the global FL model, addressing limitations seen in current FL paradigms.

**Data and Resource Heterogeneity in FL.** Data and resource heterogeneity remain significant challenges in FL, impacting both training and performance [28, 20]. Fruitful solutions have been explored to tackle the data heterogeneity [10, 25, 4, 42] or resource heterogeneity [21, 11, 7], while not in LLM context. A concurrent work, HETLORA [8], proposes allowing heterogeneous LoRA ranks by zero-padding local LoRA weights for aggregation and truncating global weights to match the local rank for distribution, all while employing sparsity regularization. However, our approach distinguishes itself through a focus on zero-shot task generalization and large-scale experiments inclusive of thousands of NLP tasks and clients, aiming to synthesize a well-generalized global LLM. Moreover, our method is simple and easy to use without any hyper-parameters for the aggregation, thereby circumventing the need for case-by-case tuning of newly introduced variables such as the decay and regularization factors of HETLORA.

## 3    Methodology of FlexLoRA

### 3.1    Intrinsic Dimension and Generalization

Fine-tuning LLMs to enhance task-specific performance inevitably encounters cost of reduced generalization ability: a trade-off supported by the "no-free-lunch" theorem and empirical evidence usually called "alignment tax" of LLM [40, 31]. The generalization capability of LLMs is influenced by complexity of applied tasks and their solution spaces, which can be characterized by the concept of an intrinsic dimension – typically far smaller than the total number of model parameters [1].

The insight of intrinsic dimension informs the design of LoRA to fine-tune LLMs' pre-trained weights in a parameter-efficient manner, utilizing compact and low-rank matrices. Specifically, matrices $A \in \mathbb{R}^{r \times p}$, $B \in \mathbb{R}^{d \times r}$ are introduced, where $r$ denotes the rank that encapsulates intrinsic dimension. These matrices form a low-rank approximation for tuning original weights $W_0$ as $h = W_0 x + sBAx$, where $x$ is the input of the parameter to be tuned, $h$ is the output, and $s$ is a scaling constant. Previous studies show that different ranks produce weights with attributes particularly tailored to specific downstream tasks [16, 18]. Consequently, the rank value plays a critical role in not only task-specific solution subspaces but also in determining a model's ability to generalize to various tasks.

In scenarios where clients have highly heterogeneous task and resource distributions, a uniform LoRA rank usually does not suffice for model performance, especially in its zero-shot generalization ability for unseen clients and tasks. Employing a small LoRA rank potentially leads to under-fitting in a global context by capturing only a subset of task-specific features, while a large rank is usually infeasible due to the "bucket effect" of existing FL solutions constrained by least-resourced clients.

FlexLoRA emerges as a solution to this dilemma by dynamically adjusting the rank in response to the variability in local client resources. By increasing the LoRA rank for clients with greater resources to contribute more global knowledge, FlexLoRA enhances the model's ability to generalize across diverse data distributions without sacrificing local performance accuracy. This strategy allows for

federated fine-tuning of LLMs to navigate between the extremes of task-specific optimization and generalization to unseen clients and tasks.

## 3.2 Aggregation with Heterogeneous Ranks

Traditional FL methods like FedAvg aggregate local LoRA weights by computing a weighted average of the decomposed matrices $A$ and $B$ as $B_g = (\sum_{i=1}^{m} n^i B_l^i)/(\sum_{i=1}^{m} n^i)$, $A_g = (\sum_{i=1}^{m} n^i A_l^i)/(\sum_{i=1}^{m} n^i)$, where $B_g, A_g$ are the global LoRA decomposed matrices, and $B_l^i, A_l^i$ are the local LoRA decomposed matrices of $i$-th client, $n^i$ is the size of the $i$-th client's local training dataset, $m$ is the number of FL clients. However, this scheme is restricted by the lowest LoRA rank among participating clients for aggregation compatibility, which makes it hard to capture the full diversity of client contributions and fully utilize ample client resources.

FlexLoRA takes a different yet simple approach to enable decomposed matrix with different LoRA ranks to be mixed together. Specifically, it first forms a low-rank approximation of the LoRA matrix for each client, $W_l^i$, before computing the weighted average: $W_g = (\sum n^i W_l^i)/(\sum_{i=1}^{m} n^i) = (\sum n^i s B_l^i A_l^i)/(\sum_{i=1}^{m} n^i)$.

After the weighted average with heterogeneous LoRA ranks, the resulting global LoRA weight $W_g$ is decomposed using SVD. Then the SVD components $U, \Sigma, V$ are redistributed to clients in a low-rank approximation that preserves as much information of $W_g$ as possible meanwhile based on clients' local resources characterized by $r^i$:

$$\text{SVD}(W_g) = U\Sigma V^T, \qquad W_g^i = U[:, :r^i]\Sigma[:r^i, :r^i]V[:r^i, :]^T \approx W_g,$$

where $U, \Sigma$, and $V^T$ are the SVD components of $W_g$, the $r^i$ within [] indicates the indexing operator of each client to select their singular vectors corresponding to top $r^i$ singular values. As a result, client $i$ receives the aggregated knowledge $W_g^i$ from server and incorporates $W_g^i$ into its local LoRA weight $W_g^i = s B_g^i A_g^i$ with $B_g^i = U[:, :r^i]\Sigma[:r^i, :r^i]/s$, and $A_g^i = V[:r^i, :]^T$.

The local training then proceeds as similar to those in standard FL approaches, using $W_l^i$ as the local weights to be tuned. This aforementioned process is repeated until convergence is achieved or a predetermined number of rounds is completed.

## 3.3 Maximizing Local Rank with Local Resources

To fully utilize the local resource of a local client, we adhere to the principle of *allocating the highest feasible rank given a client's resource budget*, which is motivated by our empirical finding that larger ranks generally yield better generalization. Figure 1 demonstrates that models trained with uniformly higher ranks outperform those with lower ranks under conventional parameter-average aggregation schemes. For single-client performances, the zero-shot performance is also boosted in the majority of the cases. The Table 13 and Figure 9 from Appendix J show that performance improves with higher LoRA ranks uniformly regardless of the tasks assigned to each client. While there might be an ideal LoRA rank that maximizes a single client's performance—potentially as high as 200—practical resource limitations may necessitate settling for a lower rank, such as 8. Therefore, we adopt the principle of setting the rank to be as large as possible to completely utilize the resources in FlexLoRA, which is easy to implement and under low risk of overfitting as Occam's razor suggests.

In Appendix B, the overall procedure of FlexLoRA and its core function of server update are summarized in Algorithm 1 and Algorithm 2 respectively. FlexLoRA optimally leverages the inherent characteristics of LoRA, boosting model generalization effectively by increasing local ranks, while without sacrificing overall training efficiency. Compared to FedAvg and homogeneous rank-based FL methods, FlexLoRA incorporates a lightweight SVD procedure, but the overhead from SVD is negligible compared to the local LLM training procedure. Moreover, the SVD is performed only once per round and is independent of client numbers. Notably, FlexLoRA enables heterogeneous ranks without the need for any additional hyperparameter tuning. As we empirically demonstrate in Table 4, the improved convergence rate, thanks to larger ranks, more than compensates for the extra overhead introduced by training on more parameters per round, resulting in a net gain in overall efficiency and a reduced overall time to completion. These features enhance its efficiency and scalability in cross-device FL settings where thousands or millions of devices are involved.

## 3.4 Generalization Analysis

We analyze the generalization ability of FlexLoRA by extending Baxter's model of learning [3]. Here, $h_W(\cdot)$ represents the hypothesis generated by the model with LoRA weights $W$, and $f\Big(W;(x,y)\Big)$ denotes the loss function for a single data point $(x,y)$. The expected loss is denoted as $\mathcal{L}(W_g) \triangleq \mathbb{E}_{(x,y)\sim\mathcal{D}_i} f\Big(W;(x,y)\Big)$. The two key assumptions underpinning our analysis are as follows:

**Assumption 1.** *The following Lipschitz conditions hold:* $|f(W;x,y) - f(W';x,y)| \leq L_f||W - W'||$ *and* $||h(W;x) - h(W';x)|| \leq L_h||W - W'||$.

Following current analysis in SOTA methodologies in LLM research, we assume Lipschitz conditions in our analysis for $f$ and $h$ [27, 17, 13]. This assumption indicates that $f$ and $h$ are Lipschitz continuous with respect to the LoRA weights $W$, ensuring the stability of the loss landscape. For simplicity, we denote $\text{SVD}(W_g, r^i)$ as using the top $r^i$ singular values and the corresponding singular vectors to approximate $W_g$. We denote the parameter solution space of $W_g$ to be $k$. Next, due to the federated average and indexing operation based on the largest singular values, we assume that the dissimilarity between the global model and its rank-constrained approximation can be bounded:

**Assumption 2.** *The LoRA weights can be bounded in a ball with radius $R$, and the error induced by the SVD approximation for each client is bounded by a constant $\phi^i$ as* $||\text{SVD}(W_g, r^i) - W_g|| \leq \phi^i$.

**Theorem 1.** *Under Assumptions 1 and 2, with probability at least $1 - \delta$, there exists a sample size $\tilde{N} = \mathcal{O}(\frac{k}{|\mathcal{C}|\epsilon^2} \log(\frac{RL_f L_h}{\epsilon - 2\phi^i L_f L_h}) - \frac{\log \delta}{|\mathcal{C}|\epsilon^2})$ such that for all $W_g$, the bound $||\mathcal{L}(W_g) - \mathcal{L}(W'_g)|| \leq \epsilon$ holds when the number of local data samples for each client $i$ exceeds $\tilde{N}$.*

Detailed proof is in Appendix C. This theorem suggests that the generalization ability of the global model is influenced by the LoRA rank $r^i$ chosen by each local client. Specifically, as $\phi^i$ is the error bound of SVD approximation, increasing rank $r^i$ makes the approximation more accurate, thus reducing $\phi^i$. Consequently, this reduction in error bound decreases the requisite number of samples, denoted as $\tilde{N}$, required for effective generalization. Moreover, an increase in the number of clients $|\mathcal{C}|$ also contributes positively to the generalization of the federated model in the order of $\mathcal{O}(\frac{1}{|\mathcal{C}|})$, a stronger impact than $\phi^i$ whose impact is in a logarithmic fashion $\mathcal{O}(log(\frac{1}{\phi^i}))$. Collectively, FlexLoRA is effective in cross-device FL settings, where the generalization capability of the global model can be significantly enhanced by the participation of massive clients (larger $|\mathcal{C}|$) with heterogeneous resources (larger $r^i$). Note that Assumption 1 is standard in FL literature [22], and our proof do not rely on simplified assumptions that often do not hold in cross-device cases, such as identically distributed data. We provide empirical support for distribution-related generalization ability, the effect of SVD (Assumption 2), and the effect of client numbers in Sections 4.3, 4.5 and 4.6 respectively.

# 4 Experiments

## 4.1 Setup for Cross-Device FL Environments

**Resource Heterogeneity.** We make FL clients resource-heterogeneous by crafting four distinct LoRA configurations as listed in Table 1. Type 1, 2, and 4 assign the same LoRA on all tunable layers, while Type 3 assigns small ranks on attention layers and large ranks on FFN layers, following the design of the MAM adapter [14]. Clients are randomly assigned a configuration type, simulating four types of heterogeneous resource environments similar to [5]. As shown in Figure 3, we consider uniform resource distribution where each LoRA configuration type is equally likely to be assigned to each client, heavy tail resource distribution where either Type 1 or Type 4 is dominant, and normal distribution where the LoRA configuration types are normal distribution and Type 2 and Type 3 are dominant. A comparison of the active memory cost with different LoRA configurations is shown in Appendix D. Besides, we add LoRA on top of all the linear layers of LLMs based on the empirical results in Appendix E.3.

**Task Heterogeneity.** We further make FL clients task-heterogeneous by utilizing the natural instruction dataset [39]. The dataset consists of over 1600 distinct natural language tasks that come from 76 NLP task types and is split based on its meta-info of the belonging NLP tasks, such that each client holds a unique task to mirror a task heterogeneous environment. Notably, while the FL setup

includes over 1600 clients, the distribution of 76 task types across these clients means that some will inherently share similar local data distributions, thereby mirroring the natural variability and overlapping task characteristics often encountered in real-world settings. Unless stated otherwise, we conduct all our FL experiments on this dataset and adopt DataJucier (1.3B) as our foundation models [6], chosen for its suitability for edge devices with constrained resources. More details about data preparation are included in Appendix E.2.

Table 1: The LoRA configurations that compose heterogeneous resource distributions, detailed in Figure 3.

| | LoRA Config | # Params |
|---|---|---|
| Type 1 | $r = 8$ on all layers | 0.12 % |
| Type 2 | $r = 30$ on all layers | 2.46 % |
| Type 3 | $r = 30$ on atten layer, $r = 200$ on FFN layer | 8.22 % |
| Type 4 | $r = 200$ on all layers | 12.22 % |

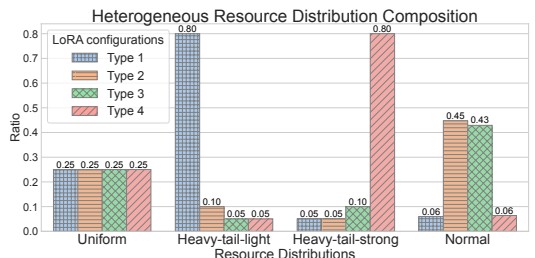

Figure 3: Heterogeneous resource distributions containing different ratios of various LoRA configuration types.

## 4.2 Setup for FL Baselines

**Baselines with Homogeneous Rank.** We adopt FedAvg [28], FedIT [43], and SLoRA [2] as baselines utilizing unvarying LoRA ranks. FedIT aggregates the LoRA module weight of each client by averaging which limits the local LoRA rank to meet the lowest resource constraint. Comparing with FedAvg, FedIT adopts Adam optimizer for local training instead of SGD optimizer. SLoRA first trains local client models with sparse fine-tuning for several epochs then switches to LoRA for PEFT and uses the updates from sparse fine-tuning as initialization.

**Baselines with Heterogeneous Ranks.** Besides, we compare with HETLORA [8], a concurrent work exploring the effective utilization of diverse LoRA ranks in FL. It first employs zero-padding on all the LoRA matrices based on the largest rank, then conducts element-wise averages like FedAvg, and finally truncates the aggregated model to fit the local client LoRA rank.

We note that both FlexLoRA and HETLORA are able to be plugged into the above-mentioned FL methods with homogeneous LoRA ranks. In our experiment, for each baseline with homogeneous rank, we also examine their performance after integration by either FlexLoRA or HETLORA.

## 4.3 Unseen Client Generalization

**Evaluation Setup.** Our initial examination focuses on the generalization capabilities of global models to unseen clients by deploying the models to clients with unseen data distributions. The unseen clients are newly sampled clients from the next communication round. This assessment allows us to measure zero-shot performance, a key indicator of a model's ability to generalize beyond the data available during the training phase. This approach is to simulate the real FL setting, where well-trained global weights will be deployed to new clients rather than clients participating in the previous round. Specifically, we investigate the performance of global models trained with baseline methods both with and without the integration of FlexLoRA and HETLORA under four distinct resource heterogeneity scenarios.

**Overview Performance Comparison.** Table 2 displays the zero-shot performance under Rouge-L scores on the test set from unseen clients, facilitating a comparison of the generalization capabilities across different federated global models. It is observed from the table that in most of the cases, methods with heterogeneous LoRA ranks have better performance than that of homogeneous ranks, indicating that heterogeneous LoRA ranks enhance the clients with larger ranks to fully exploit their capability. Furthermore, among the heterogeneous LoRA rank methods, our proposed FlexLoRA consistently outperforms HETLORA across all resource distribution settings. This shows that FlexLoRA is able to take advantage of heterogeneous resource distribution, and is more capable of leveraging the general information from heterogeneous LoRA configurations.

Table 2: The weighted average Rouge-L scores of unseen clients provide insights into the global model's generalization ability. Results from baseline methods with homogeneous ranks (Line 3, denoted as Homo Rank) are compared with those incorporating FlexLoRA and HETLORA across various resource distributions (Line 4∼7). The significant test are presented in Appendix F.

| | FedAvg | | FedIT | | SLoRA | |
|---|---|---|---|---|---|---|
| | FlexLoRA | HETLORA | FlexLoRA | HETLORA | FlexLoRA | HETLORA |
| Homo Rank | $56.53_{\pm 0.17}$ | | $61.29_{\pm 0.93}$ | | $60.01_{\pm 0.74}$ | |
| Uniform | $\mathbf{58.07}_{\pm 0.27}$ | $56.85_{\pm 0.18}$ | $\mathbf{61.34}_{\pm 1.09}$ | $60.74_{\pm 0.78}$ | $\mathbf{60.75}_{\pm 0.60}$ | $60.74_{\pm 0.77}$ |
| Heavy-Tail-Light | $\mathbf{57.39}_{\pm 0.54}$ | $56.24_{\pm 0.30}$ | $\mathbf{61.88}_{\pm 0.89}$ | $61.53_{\pm 0.93}$ | $\mathbf{60.40}_{\pm 0.40}$ | $59.97_{\pm 0.62}$ |
| Normal | $\mathbf{57.78}_{\pm 0.33}$ | $56.50_{\pm 0.05}$ | $\mathbf{62.01}_{\pm 0.91}$ | $61.03_{\pm 0.54}$ | $\mathbf{61.67}_{\pm 1.07}$ | $61.14_{\pm 0.71}$ |
| Heavy-Tail-Strong | $\mathbf{57.73}_{\pm 0.08}$ | $55.74_{\pm 0.93}$ | $\mathbf{62.20}_{\pm 1.12}$ | $61.06_{\pm 0.95}$ | $\mathbf{61.86}_{\pm 1.24}$ | $61.29_{\pm 0.95}$ |

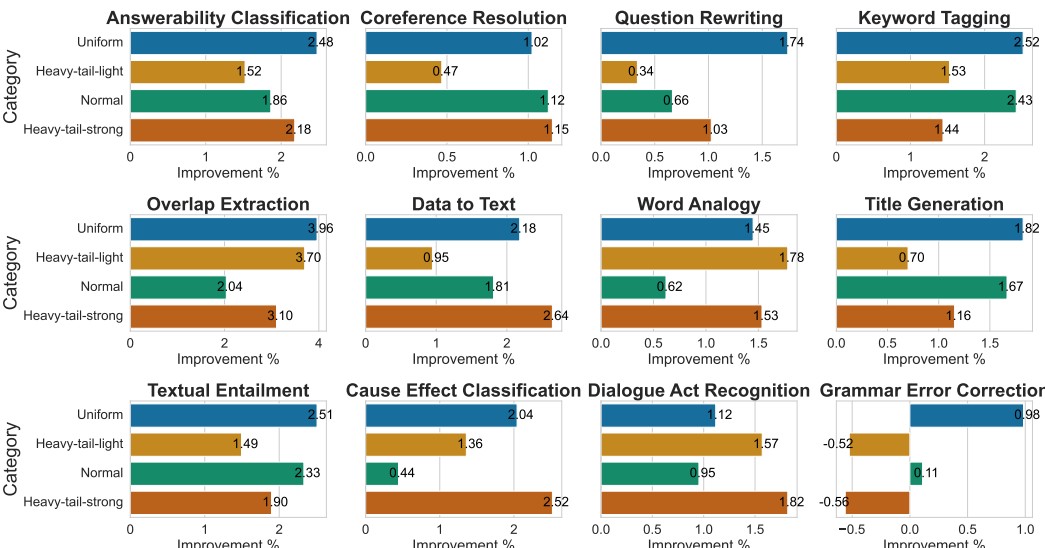

Figure 4: Task-specific improvements achieved by FlexLoRA in comparison with the homogeneous rank implementation of FedAvg, across different resource distribution settings.

**Effect of Specific Resource Distributions.** To gain further insight into the effect of FlexLoRA and the effect of heterogeneity of LoRA ranks, in Table 10 in Appendix F, we list the percentage improvement for each FL methods when incorporating FlexLoRA in comparison with the respective standard homogeneous rank implementations. Notably, after integrating FlexLoRA, the average performance gains are 2.14% for FedAvg, 0.86% for FedIT, and 1.94% for SLoRA. These enhancements lend empirical support to our theoretical generalization analysis that clients utilizing higher LoRA ranks tend to exhibit improved generalization abilities. The most substantial performance improvements are observed in the heavy-tail-strong resource distribution, followed by the normal distribution. This is consistent with our expectations since the heavy-tail-strong distribution predominantly comprises clients with Type 4 LoRA configurations (rank 200). The limited presence of Type 1 clients (rank 8) in the normal distribution minimizes the risk of the global model being excessively influenced by task-specific LoRA weights. Therefore, FlexLoRA is able to leverage the heterogeneous resource distributions to boost the zero-shot generalization, and the gain from integrating FlexLoRA is directly related to the ratio of clients with heavy resources.

## 4.4 Cross-Task Generalization

**Evaluation Setup.** To assess the natural language understanding capabilities of the FlexLoRA-enhanced global model, we evaluate its performance on a range of downstream NLP tasks. Specifically, the model is tested on the English Track of the evaluation tasks from [39], featuring 12 categories and 119 tasks. For each task, a random sample of 100 data points is chosen for testing. We

Table 3: Average percentage improvement of FlexLoRA over baseline methods (FedAvg, FedIT, SLoRA) across different resource distributions, calculated over 12 NLP task categories. More detailed comparison is presented in Figure 4.

|  | FedAvg | FedIT | SLoRA | Avg |
| --- | --- | --- | --- | --- |
| Uniform | 1.99% | 0.97% | 0.74% | 1.23% |
| Heavy-Tail (L) | 1.24% | 0.63% | -0.47% | 0.47% |
| Normal | 1.34% | 0.75% | 0.96% | 1.02% |
| Heavy-Tail (S) | 1.66% | 0.95% | 1.12% | 1.24% |
| Avg | 1.56% | 0.83% | 0.59% | 1.00% |

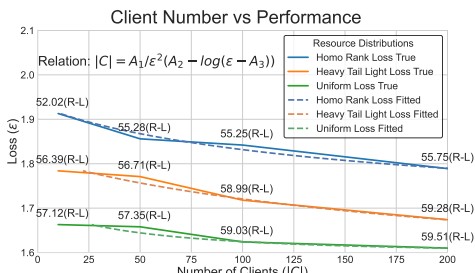

Figure 5: Client number ($|\mathcal{C}|$) v.s. generalization on heterogeneous distributions. Increasing $|\mathcal{C}|$ improves both convergence loss and rouge-L (scores marked as (R-L)).

summarize the average percentage improvement achieved by integrating FlexLoRA across different resource distributions in Table 3.

**Effect of Specific Resource Distributions.** In the majority of cases, the global models augmented with FlexLoRA demonstrate marked improvements over the vanilla implementations of FedAvg, FedIT, and SLoRA by up to 1.99%, suggesting that the FlexLoRA also improves the natural language analysis capabilities. An exception is noted in the SLoRA on the heavy-tail-light distribution, potentially due to the predominance of the Type 1 LoRA configuration (rank 8), which may limit the overall language processing capabilities when such clients are disproportionately represented. This configuration's minimal rank assignment across all linear layers suggests that the local weight aggregation on the server side might not fully leverage the capabilities of clients with larger resources, potentially detracting from the model's performance on certain tasks.

**Effect of Specific Language Tasks.** We further illustrate the task-specific improvements of integrating FlexLoRA in comparison with the standard FedAvg configuration for various resource distributions in Figure 4. The task-wise improvement figures for other FL methods are included in Appendix G. We observe that the global models trained using the FlexLoRA aggregation scheme generally outperform others on tasks requiring the parsing of logical relationships between sentences. Particularly, it gains improvements at most 4% in the overlap extraction task, and around 2.5% in the textual entailment, cause-effect classification, and dialogue act recognition task, verifying again the effectiveness of FlexLoRA.

## 4.5 Aggregation Scheme Study

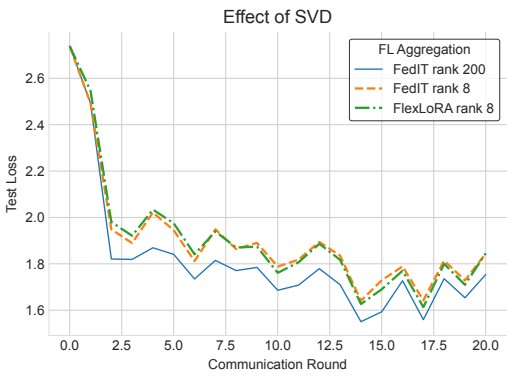

(a) Losses of FedIT and FlexLoRA-integrated FedIT.

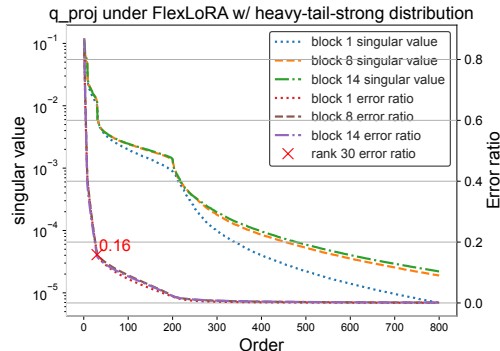

(b) The $q_{proj}$'s singular values and estimate error.

Figure 6: The sub-figure 6(a) shows that FedIT with LoRA rank 8 has comparable test loss curves for standard and FlexLoRA integration. At rank 200, though, standard FedIT differs from other versions. 6(b) depicts singular value distributions and approximation errors, where the red cross indicates the average error for rank 30 $q_{proj}$ weights in specific blocks. Further details are in Appendix H.

**SVD on Convergence.** FlexLoRA's aggregation scheme constructs full-size LoRA weights before averaging, unlike the conventional FedAvg method's parameter-wise averaging. To understand the effect of this difference on model performance, we assess the performance of FlexLoRA in a controlled environment using homogeneous LoRA ranks and compare it to the standard FL aggregation scheme. Figure 6(a) presents the test loss trajectories for FedIT with and without the FlexLoRA enhancement, both utilizing a homogeneous rank of 8. The loss curves for the standard FedIT and FedIT with FlexLoRA closely align, suggesting comparable performance. In a nutshell, we empirically demonstrate that under homogeneous conditions, FlexLoRA's aggregation does not negatively impact model performance compared to traditional methods. Besides, it's worth noting that the test loss for FedIT with a homogeneous rank of 200 is significantly lower, underscoring the benefits of higher rank configurations and evidencing our Theorem 1.

**SVD v.s. LoRA Rank.** To gain further insight into the effect of SVD, we calculate and sort the singular values of the global LoRA weights from largest to smallest. We focus on specific layers where LoRA is applied within the transformer blocks 1, 8, and 14 (each block includes one attention layer and one FFN layer). Figure 6(b) displays the scale of singular values and the error ratio between the global LoRA weights approximated by the top $i$ singular values and the full-rank global LoRA weights in the FedIT setting with a heavy-tail-strong resource distribution. The approximation error is quantified as the norm of the difference between the approximated and full-rank weights. The error curves for $q_{proj}$ layers across all transformer blocks nearly overlap. With the weight approximated from the top 30 ranks, the error ratio is as low as 0.16, suggesting the approximated weights are in close proximity to the actual full-rank weights, lending empirical support to our Assumption 2.

## 4.6 Scalability Study

**Larger Model Size and Lower Degree of Task Heterogeneity.** While the aforementioned experiments with 1.3B LLM and meta-task dataset split effectively showcases FlexLoRA's capabilities against baselines in a highly heterogeneous cross-device environment, real-world settings may be relaxed involving fewer clients with a mixture of tasks and larger LLM. To better evaluate FlexLoRA's performance in such scenarios, we expanded our study to include settings where each client manages not just one specific "meta" task but a variety of different tasks. We use Dolly-15K dataset [9], which supports instruction tuning and includes 8 tasks in total. We distributed this dataset among 200 clients using a Dirichlet distribution with $\alpha = 0.5$ to simulate the non-IID data distributions. Moreover, we also incorporate the most cutting-edge LLaMA-3 [37] with 8B parameter size as our foundation model to assess FlexLoRA's effectiveness with advanced, larger-scale models.

Table 12 in Appendix I illustrates FlexLoRA's efficacy under both smaller and larger foundation models within the mixture of task settings. Compared with LLaMA-3 (8B), finetuning on DataJucier(1.3B) demonstrates more generalization improvement for unseen clients. This enhanced performance when using the smaller DataJuicer model suggests that FlexLoRA is particularly effective for foundation models that are scalable to edge devices. Such ability is instrumental in maximizing the utility and efficiency of smaller models in resource-constrained environments.

**System Costs.** We empirically demonstrate that increasing the rank improves overall efficiency. The experiment is conducted on the Dolly 15K dataset and the DataJuicer 1.3B model, with the same FL setting as Table 12. We summarize the results in Table 4, where the "$R$" indicates the number of rounds to reach a loss of 2, approximately 75% progress to convergence. The "$Cost_R$" stands for per-round FL cost in terms of the average model parameters compared with the foundation model parameters, as both

Table 4: Convergence round and FL cost per round for different LoRA ranks.

|  | $R$ | $Cost_R$ | $Cost_{all}$ |
|---|---|---|---|
| Homo Rank | 48 | 1.001x | $\approx 100\%$ |
| Heavy-Tail (L) | 24 | 1.014x | $\approx 50.6\%$ |
| Uniform | 15 | 1.061x | $\approx 33.1\%$ |

the local training and communication cost positively correlated to the rank value in FlexLoRA (as analyzed in Sec. 3.3). The "$Cost_{all}$" indicates total cost calculated as multiply of $R$ and $Cost_R$, and by setting the result of "Homo Rank" as the baseline. From the results, we can see that FlexLoRA achieves faster convergence with slightly increased parameter percentages under heterogeneous resource distributions. For the overall efficiency, "Heavy-Tail-Light" has a total reduction of 49.4%, and "Uniform" has 66.9%, indicating good scalability of FlexLoRA.

**Effect of Larger Client Number.** We designed an experiment to empirically validate Theorem 1 and demonstrate how FlexLoRA performs with varied client numbers. We use subsets of 10, 50, and 100 clients from a pool of 200 clients on Dolly-15K dataset and DataJuicer (1.3B), with FedAvg as baseline FL method and each FL round always sampling 10 participants. From Figure 5, there is a marked improvement in generalization as the number of participating clients increases. Moreover, from Theorem 1, we can derive the functional relationship between client number $|C|$ and the generalization loss $\epsilon$ as $|C| = A_1/\epsilon^2 (A_2 - log(\epsilon - A_3))$, where $A_{1,2,3}$ are constants absorbing the other factors impacted by specific resource distribution in this experiment. We thus fit these coefficients using the derived form and empirical observations, and find that the dotted curves gain good fitness for all the tested distributions. There results affirm the preciseness of our Theorem 1 again, indicating the suitableness of FlexLoRA for cross-device FL scenarios where leveraging a broad client pool can boost the generalization across diverse data distributions.

## 5  Conclusion

In this work, we propose a simple yet effective method named FlexLoRA to address the challenges posed by resource and data heterogeneity among clients during the federated fine-tuning of LLMs. By leveraging larger local LoRA ranks, FlexLoRA not only improves the generalization ability of the global model but also ensures that all clients, irrespective of their resource capabilities, can contribute meaningfully. Theoretical analysis and extensive experiments verify the effectiveness and scalability of FlexLoRA. Due to resource limitations, we have not tested the LLaMA-3 model on thousands-client scenarios, which we leave as future work. We hope this study can enlighten more future research and development in data-efficient and privacy-preserving enhancement of LLMs.

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

## Appendix

We provide an outline of how we organize the appendix in below:

**Implementation details** of our experiments:

- Appendix A: Implementation details for the experiments shown in Figure 1 related to the zero-shot test loss of naive FedIT with LoRA rank of 1, 8, and 200, including an "extreme heavy tail" scenario.
- Appendix E: General experimental details, including hyperparameter settings, cross-task splitter details, and the choice of layers for applying LoRA.
- Appendix L:Provides details about the computational infrastructure and cost associated with the experiments mentioned in the document.

**Algorithm, proof and analysis** of FlexLoRA:

- Appendix B: The pseudocode for FlexLoRA's algorithm in a federated learning context, detailing the process from initialization, client sampling, local updates, to server updates.
- Appendix C: Proof of Theorem 1, providing a mathematical derivation of the bounds on generalization error under specific assumptions.
- Appendix D: Analysis of efficiency improvements from LoRA under different ranks, detailing the trainable parameters, memory costs, and efficiency gains compared with full finetuning.

More **experimental results** including:

- Appendix F: Statistics of the Table 2, including 1. Percentage of improvement w/ FlexLoRA vs w/o FlexLoRA 2. significant test results comparing FlexLoRA with its baselines across different resource distribution types.
- Appendix G: More results on task-specific performance improvements of global models trained with FlexLoRA, particularly in natural language tasks.
- Appendix H: Additional results on the effect of SVD, including the distribution of singular values and the approximation error ratio.
- Appendix I: Results on the efficacy of FlexLoRA result under a mixture of task setting.
- Appendix J: Discusses the single client's performance impact of different LoRA ranks, showcasing how performance improves with higher LoRA ranks across various NLP tasks.

## A  Implementation of Figure 1

For the experiments in Figure 1, we plotted the zero-shot test loss of naive FedIT with LoRA rank equal to 1, 8, and 200. For the experiment of FedIT with FlexLoRA, we adopt an "extreme heavy tail" scenario where all the clients have a LoRA rank of 200 except one with a LoRA rank of 8. All the hyperparameters and FL experiment settings are the same as the experiments in Table 2.

## B  The Algorithm of FlexLoRA

We summarize the Pseudocode of FlexLoRA in Algorithms 1 and 2.

## C  Proof of Theorem 1

Let $\mathbb{H}^n$ denote the function space with its elements parameterized by $W_g$ and the distance metric $\Delta$ is defined as:

$$
\begin{aligned}
&\Delta(W_g^i - W_g^{i'}) \\
&= \frac{1}{n} \mathbb{E}_{x,y \sim \mathcal{D}_i} \left[ \left| \sum (f(W_g^i; x, y) - \sum f(W_g^{i'}; x, y) \right| \right],
\end{aligned}
\tag{1}
$$

---

**Algorithm 1** FlexLoRA for Federated Learning

---

**Input:** $T$, $B_g^0$, $A_g^0$, $\{D_i^0\}_{i \in \mathcal{C}}$, $s$, $r_g$, $\{r^i\}_{i \in \mathcal{C}}$, where $T$ is the total communication rounds and $\mathcal{C}$ is the set of all FL clients

1 **for** $t = 1, \cdots, T$ **do**
2    Server samples clients $\mathcal{C}^t$ sampled from $\mathcal{C}$, sends $B_g^i$ and $A_g^i$ to $i \in \mathcal{C}^t$
    **for** *client $i \in \mathcal{C}^t$ in parallel* **do**
3       Update local LoRA module weight:
      $\{B_l^i, A_l^i\}_{i \in \mathcal{C}} = \text{LOCALUPDATE}\left(\{B_g^i A_g^i\}_{i \in \mathcal{C}}\right)$
4    Server updates global model:
    /* Implement FlexLoRA Here */
    $\{B_g^i, A_g^i\}_{i \in \mathcal{C}} = \text{SERVERUPDATE}\left(\{B_l^i, A_l^i, r^i\}_{i \in \mathcal{C}}\right)$
5 **return** $\theta$, $\{B_g^i, A_g^i\}_{i \in \mathcal{C}}$

---

---

**Algorithm 2** FlexLoRA Server Update

---

**Input:** $\{B_l^i, A_l^i, r^i, \gamma^i\}_{i \in \mathcal{C}}$, $s$, $r_g$, $\{r^i\}_{i \in \mathcal{C}}$, where $\gamma^i$ is average constant for client $i$ depending on size of its local dataset

6 *Aggregation with Heterogeneous LoRA Ranks*:
   Initialize global weight $W_g = 0$
   **for** $i \in \mathcal{C}^t l$ **do**
7     Compose local client LoRA weight:
    $W_l^i = s B_l^i A_l^i$
    Weighted Average client weight:
    $W_g = W_g + \gamma^i W_l^i$
8 *Decompose global LoRA weight*:
  /* Only do once */
  $U, \Sigma, V = \text{SVD}(W_g)$
  *Distribute Global Weight*:
  **for** $i \in \mathcal{C}^t l$ **do**
9     Compute each client LoRA weight based on their resource limitation:
    $B_g^i = U[:, : r^i] \Sigma[: r^i, : r^i] / s$
    $A_g^i = V[: r^i, :]^T$
10 **return** $\{B_g^i, A_g^i\}_{i \in \mathcal{C}}$

---

With the inequalities introduced by Assumptions 1 and 2, we have:

$$
\begin{aligned}
&\Delta(W_g^i - W_g^{i,'}) \\
&= \frac{1}{n} \mathbb{E}_{x,y \sim \mathcal{D}_i}\left[\left|\sum(f(W_g^i; x, y) - \sum f(W_g^{i,'}; x, y)\right|\right] \\
&\leq L_f \|h_{W_g^i} - h_{W_g^{i,'}}\| \\
&\leq L_f L_h \|\text{SVD}(W_g, r^i) - \text{SVD}(W_g', r^i)\| \\
&= L_f L_h \|\text{SVD}(W_g, r^i) - W_g - \text{SVD}(W_g', r^i) \\
&\quad + W_g' + W_g - W_g'\| \\
&\leq L_f L_h \|\text{SVD}(W_g, r^i) - W_g\| \\
&\quad + L_f L_h \|\text{SVD}(W_g', r^i) - W_g'\| \\
&\quad + L_f L_h \|W_g - W_g'\| \\
&\leq L_f L_h \left[2\phi^i + \|W_g - W_g'\|\right].
\end{aligned}
\tag{2}
$$

Let the parameter solution space of $W_g$ denote as $k$. We can get an $\epsilon$-covering in metric $\Delta(W_{g,i} - W_{g,i}')$ if we select a covering in the parameter space with $\|W_g - W_g'\|$ equal to $\frac{\epsilon}{L_f L_h} - 2\phi^i$. Therefore, the covering number of $\mathbb{H}^{|\mathcal{C}|}$, denoted as $\mathcal{B}(\epsilon, \mathbb{H}^{|\mathcal{C}|})$ is: $\log\left(\mathcal{B}(\epsilon, \mathbb{H}^{|\mathcal{C}|})\right) = \mathcal{O}\left(k \log(\frac{R L_f L_h}{\epsilon - 2\phi^i L_f L_h})\right)$.

According to previous studies [3, 35, 7], there exists $\tilde{N}$ and $\tilde{N} = \mathcal{O}\left(\frac{1}{n\epsilon^2}\log\frac{\mathcal{B}(\epsilon,\mathbb{H}^{|\mathcal{C}|})}{\delta}\right) = \mathcal{O}(\frac{k}{|\mathcal{C}|\epsilon^2}\log(\frac{RL_fL_h}{\epsilon-2\phi^iL_fL_h}) - \frac{\log\delta}{|\mathcal{C}|\epsilon^2})$. □

## D  Efficiency Improvement from LoRA under Different Ranks

We calculate the % trainable parameters and active memory cost for LoRA under different ranks compared with full finetuning in our experiment setting, summarized in Table 5. Although the size of LoRA weights is small compared with the pretrained model because LoRA only tunes a small proportion of the trainable parameters, LoRA tuning is still much more efficient than full parameter finetuning, which highlights the necessity to scale LoRA rank size on clients with diverse computation resources. Table 6 illustrates the empirical time consumption for tuning client with LoRA and full finetuning.

Table 5: Trainable parameters and memory cost for different LoRA configurations, and the corresponding efficiency improvement compared with full finetuning.

|  | LoRA Config | % Trainable Parameters | Memory Cost | Improvement |
|---|---|---|---|---|
| Type 1 | $r = 8$ on all layers | 0.12 % | 34.71GB | 833.3x / 1.83x |
| Type 2 | $r = 30$ on all layers | 2.46 % | 38.70GB | 40.7x / 1.64x |
| Type 3 | $r = 30$ on atten layer, $r = 200$ on FFN layer | 8.22 % | 42.57GB | 12.2x / 1.49x |
| Type 4 | $r = 200$ on all layers | 12.22 % | 46.54GB | 8.2x / 1.37x |
| Full finetuning | N/A | 100 % | 63.62GB | 1.0x / 1.0x |

Table 6: Speedup of LoRA tuning for each communication round.

| LoRA Rank (homo) | Average time per client | Speedup |
|---|---|---|
| r=8 | 1 min 17 sec | 2.31x |
| r=200 | 1 min 41 sec | 1.76x |
| Full finetuning | 2 min 57 sec | 1.0x |

Table 7: Illustration of the original data structure for tasks in natural instruction dataset [39].

| Task Type | Cause Effect Classification |
|---|---|
| Task ID | task828_copa_cause_effect_classification |
| Definition | In this task your given two statements. You must judge whether the second sentence is the cause or effect of the first one. Label the instances as "cause" or "effect" based on your judgment. The sentences are separated by a newline character. |
| Positive Example | **Input**: The women met for coffee. They wanted to catch up with each other. **Output**: cause **Explanation**: The women met for coffee because they wanted to catch up with each other. |
| Negative Example | **Input**: My body cast a shadow over the grass. The sun was rising. **Output**: effect **Explanation**: The rising of the sun isn't an effect of casting a shadow over the grass. |
| Instance | **Input**: The woman tolerated her friend's difficult behavior. The woman knew her friend was going through a hard time. **Valid Output**: ["cause"] |

## E  Implementation Details

### E.1  Hyperparameter Setting

All FL experiments are conducted with a client participation rate of 0.05 in each round, and with an early stopping mechanism that terminates training if the validation loss does not improve over 3 consecutive FL rounds. These values are adopted to better reflect the operational challenges inherent in real-world cross-device scenarios, which often involve limited device responsiveness and restricted

Table 8: The example of prompt template for training, adapting from Table 7.

| Instruction | In this task your given two statements. You must judge whether the second sentence is the cause or effect of the first one. Label the instances as "cause" or "effect" based on your judgment. The sentences are separated by a newline character. |
|---|---|
| Input | The woman tolerated her friend's difficult behavior. The woman knew her friend was going through a hard time. |
| Output | "cause" |
| Category | Cause Effect Classification |

training duration. Besides, the batch size is set as 4 via searching from a range of {2,4,16}. The maximum token length is 512. All the experiments are repeated with 2 random seeds and we report the standard deviations. More details about the computational infrastructure and cost for our experiments are in Appendix L.

For the sparse fine-tuning stage of SLoRA, we set its sparsity corresponding with the resource cost of the client's LoRA configuration shown in Table1. For example, if the client is assigned with Type 1 LoRA configuration, we will generate a mask with a sparsity of 0.12%. For HETLORA, following the original paper, we adopt 0.99 as the decay factor for rank pruning and search the regularization factor from a range of {5e-2, 5e-3, 5e-4}. For both the experiments integrated with and without FlexLoRA, we grid search their learning rates from a range of {5e-2, 5e-3, 5e-4} for FedAvg, and {5e-4, 1e-4, 5e-5, 1e-5} for FedIT and SLoRA, both accompanied with a linear scheduler which decays from the initial learning rate to 0.

### E.2 Cross-Task Splitter Details

Echoing findings from [39] on model performance saturation with few data instances, we randomly sample 10% of each task's data to scale experiments. For all the datasets we used, data for each client is partitioned into training, validation, and testing sets in a ratio of 8:1:1. In the initial setup of the natural instruction dataset, each task is accompanied by its definition, along with a positive example and a negative example. To adapt this dataset for model fine-tuning through instruction tuning, we transform the structure so that the task definition serves as the direct instruction for each data instance. This restructuring is illustrated by comparing the original and modified data formats in the natural instruction dataset, as shown in Table 7 and Table 8. This adjustment ensures that each instance is now explicitly aligned with its instructional context, facilitating a more straightforward and effective fine-tuning process.

### E.3 Choice of Layers for applying LoRA

We explore the influence of insertion layers of LoRA on the performance of the fine-tuning. While [16] and [8] add LoRA on all the attention layers, [43] adds LoRA on all the linear layers, i.e., the attention layers and Feed-Forward Network layers. We experimented with both approaches to adding LoRA. From Table 9, we demonstrate that adding LoRA to both the attention layers and Feed-Forward Network layers boosts generalization performance and adopt this setting in our experiments.

Table 9: Impact of choosing different layers to apply LoRA module. The results are zero-shot Rouge-L score of the global model. For the experiment that uses FlexLoRA for aggregation, the client resource distribution is uniform.

| Experiment | Atten Layers | All layers |
|---|---|---|
| FedIT | 0.5701 | 0.6195 |
| w/ FlexLoRA | 0.5751 | 0.6211 |

## F Relative Improvement for Table 2 and Significant Test

Table 10 presents the percentage of improvement for Table 2, providing a more straightforward overview of the enhancement of FlexLoRA.

Table 10: Percentage of improvement of FedAvg, FedIT, and SLoRA incorporating with FlexLoRA compared with their respective configurations without FlexLoRA, as shown in Table 2.

|  | FedAvg | FedIT | SLoRA | Avg |
|---|---|---|---|---|
| Uniform | 2.72% | 0.08% | 1.22% | 1.33% |
| Heavy-Tail-Light | 1.52% | 0.97% | 0.65% | 1.05% |
| Normal | 2.20% | 1.17% | 2.78% | 2.05% |
| Heavy-Tail-Strong | 2.12% | 1.24% | 3.09% | 2.15% |
| Avg | 2.14% | 0.86% | 1.94% | 1.65% |

The statistical test results presented in Table 11 provide a quantitative comparison between the performance of FlexLoRA across different FL baselines to be integrated, namely FedAvg, FedIT, and SLoRA. The p-values obtained from the statistical tests are used to determine whether the observed differences in performance are statistically significant.

Table 11: Significant test results (in p-values) between FlexLoRA and its FL baselines to be integrated.

|  | **FedAvg** | **FedIT** | **SLoRA** |
|---|---|---|---|
| Uniform | 0.064 | 0.358 | 0.044 |
| Heavy-Tail-Light | 0.168 | 0.018 | 0.175 |
| Heavy-Tail-Strong | 0.016 | <0.001 | 0.060 |
| Normal | 0.029 | 0.006 | 0.044 |

As shown in the table, FlexLoRA consistently outperforms its homogeneous baselines under various distribution types, indicated by p-values smaller than the conventional significance level of 0.05 in several cases. Notably, under the 'Heavy Tail Light' distribution, FlexLoRA achieves statistically significant improvements in FedIT (p=0.018) scenarios. Highly significant improvements are also observed in the "Heavy Tail Strong" and "normal" distributions, especially in the FedIT setting, where the p-values are smaller than 0.001 and 0.006 respectively, strongly suggesting that FlexLoRA's performance enhancement is not due to random chance.

Overall, the results indicate that FlexLoRA is a robust approach that can yield performance improvements in federated tuning environments for LLMs, especially in scenarios characterized by a resource-heterogeneous distribution of client resources.

## G   Natural Language Task Performance

Figure 7 illustrates the performance of global models on natural language tasks when trained with FlexLoRA in the FedIT setting. In line with results from the FedAvg setting, global models that leverage heterogeneous ranks in the FedIT setting also perform better on tasks involving logical relationships between sentences, such as overlap extraction, textual entailment, cause-effect classification, and dialogue act recognition. However, the global models underperform in word-level analysis tasks, such as word analogy and keyword tagging. We hypothesize that this underperformance in the FedIT setting is due to the local clients being trained with the Adam optimizer rather than the conventional SGD optimizer. The top decoding block of the LLaMA transformer, which processes word-level information, likely overfits specific information because of the momentum component inherent to the Adam optimizer.

## H   More Results for Effect of SVD

Similar to Figure 6(b), Figure 8 also verifies the empirical information loss from SVD by illustrating the singular value distribution and error ratio of $k_{proj}$ weights. $k_{proj}$ weights shares similar trends with $q_{proj}$ weights regarding the singular value distribution and error ratio.

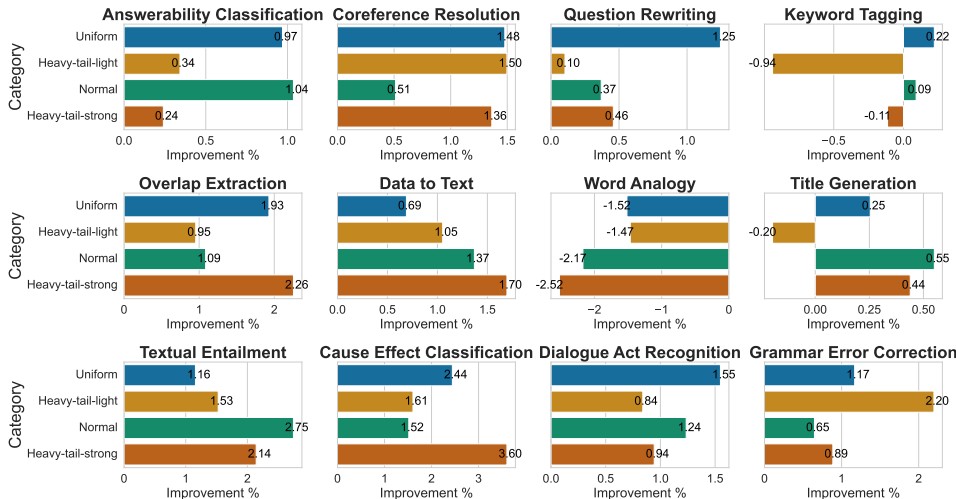

Figure 7: Task-specific improvements achieved by FlexLoRA in comparison to the homogeneous rank implementation of FedIT, across different resource distribution settings.

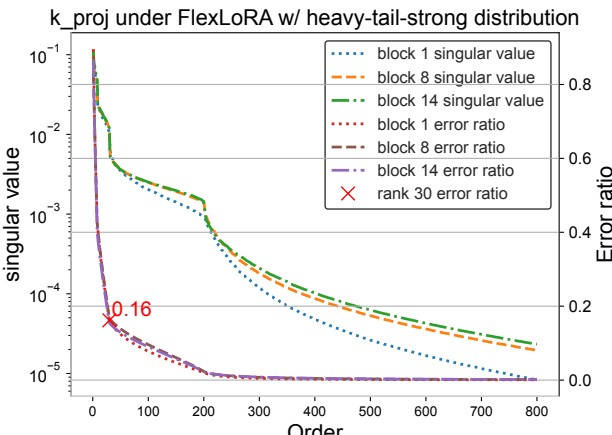

Figure 8: Distribution of singular values and the approximation error ratio between the top $i$-th singular value approximated $k_{proj}$ weights and the actual full-rank $k_{proj}$ weights. The red cross denotes the average error for weights with rank 30 of $k_{proj}$ across blocks 1, 8, and 14.

# I   Experiment Result for Mixed Task Heterogeneity Scenario

Table 12 provides the result of standard FedAvg vs FlexLoRA-integrated FedAvg under uniform and heavy-tail-light resource distributions. The foundation model includes DataJucier(1.3B) and LLaMA 3 (8B).

Table 12: Results of homogeneous LoRA configurations versus FlexLoRA under FedAvg methods. The experiments are conducted on both DataJucier (1.3B) and LLaMA-3(8B) models on Dolly-15K.

|  | DataJucier (1.3B) | LLaMA-3 (8B) |
|---|---|---|
| Homo Rank | $55.34 \pm 0.59$ | $64.26 \pm 0.80$ |
| Uniform | $58.56 \pm 1.35$ | $64.93 \pm 0.33$ |
| Heavy-tail-light | $58.39 \pm 1.26$ | $64.58 \pm 0.62$ |

## J  Single Client's Performance under Different Rank

Table 13 below shows the single client performance under a small rank of 8 and a large rank of 200. This data shows that performance improves with higher LoRA ranks uniformly regardless of the tasks assigned to each client, motivating us to allocate the highest feasible rank given a client's resource budget. Figure 9 illustrates all the clients' performance under rank 8 and rank 200.

Table 13: 8 examples of single-client performance under FedIT with homo rank 8 and homo rank 200 distribution in a single round.

| rank=8 | rank=200 | Task Type |
|--------|----------|-----------|
| 0.8977 | 0.9157 | Translation |
| 0.9084 | 0.9272 | Translation |
| 0.549 | 0.5749 | Program Execution |
| 0.5031 | 0.5301 | Program Execution |
| 0.4414 | 0.4507 | Sentiment Analysis |
| 0.5173 | 0.5328 | Fact Verification |
| 0.6995 | 0.7354 | Program Execution |
| 0.4801 | 0.5166 | Question Rewriting |

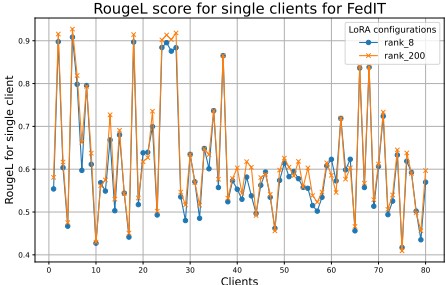

Figure 9: Performance on FedIT with homo rank 8 vs homo rank 200 across all the clients.

## K  FlexLoRA with Centralized LoRA Adaptive Methodologies

Besides directly applying LoRA to each clients, we also investigate the efficacy of existing dynamic LoRA methods in central learning. We adapts concept of ReLoRA[24] into our setting and explore its utility when combining with FlexLoRA. ReLoRA trains LoRA for several epochs, uploads the LoRA weight to the pretrained weights, then initializes a new LoRA module and trains based on the new frozen weights, repeating this process for several LoRA modules. We create a baseline by incorporating the concept of ReLoRA, uploading aggregated LoRA weights to the pretrained models after several communication rounds.

However, we observed a slower convergence speed than the regular FlexLoRA. We show in Table 14 below that incorporating the step of "uploading LoRA weights" does not lead to better performance. This suggests that observations and improvements seen in centralized dynamic LoRA methods may not translate directly to federated learning due to the unique challenges posed by heterogeneity in FL, which underlines the importance of tailored solutions like FlexLoRA for effectively managing heterogeneity in FL environments.

Table 14: Results with/without incorporating ReLoRA into either regular FedAvg or FlexLoRA.

| | Homo Rank (Baseline) | Uniform |
|---|---|---|
| Table 2 Result | 56.53 | 58.07 |
| +ReLoRA Result | 53.9 | 56.56 |

## L   Computational Resources and Infrastructure Report

In our study, we employ the LLaMA-1.3B from Data-Juicer [2] as the foundational architecture for our FlexLoRA framework, which consists of approximately 1.35 billion parameters. During our experiments, the DataJuicer-1.3B model itself is kept frozen, and the tunable parameter size for each client within our federated learning framework are determined by the size of the LoRA module, with specific configurations detailed in Table 1. Our experiments are conducted on a cluster equipped with 16 NVIDIA A100 GPUs, each with 40GB or 80GB of memory. The total GPU hours for running all the experiments is approximately 1200 GPU hours. All the experiments are implemented using PyTorch package with version 2.1.0 and Huggingface's Transformers package with version 4.31.0.

---

[2] https://huggingface.co/datajuicer/LLaMA-1B-dj-refine-150B

