# OpenReview forum: "Federated Fine-tuning of Large Language Models under Heterogeneous Tasks and Client Resources"
_NeurIPS.cc/2024/Conference — NeurIPS 2024 poster_

### Official Review · Reviewer_ug2H · 2024-07-02

**Soundness:** 3
**Presentation:** 2
**Contribution:** 3
**Rating:** 5
**Confidence:** 4

**Summary:**

The use of LoRA in FL is challenged by the heterogeneity of downstream tasks and available resources among clients. Traditional FL methods often use the smallest viable LoRA rank for all clients for aggregation compatibility, which makes it hard to capture the full diversity of client contributions and fully utilize ample client resources. To fully leverage heterogeneous client resources for enhancing the global model's generalization ability, this paper proposes FlexLoRA, which enables the mixture of diverse LoRA weights across individual clients to account for local client resources and task differences by allocating the highest feasible rank given a client's resource budget to ensure all clients contribute effectively regardless of resource capability, with heterogeneous aggregation and redistribution of weights through SVD. Extensive experiments and theoretical analysis verify the effectiveness and scalability of the proposed FlexLoRA.

**Strengths:**

1.	Exploring heterogeneous rather than homogeneous LoRA ranks across clients to fully utilize their resources is an imperative research direction.
2.	Numerous experiments and theoretical analyses are illustrated to support the effectiveness of the proposed method.

**Weaknesses:**

1.	Numerous typos remain in this paper, including but not limited to: 1) "${Client_j}$" should be corrected to "${Client&nbsp;j}$" in Figure 2; 2) "Effect" should be corrected to "effect" in line 249; 3) "r=200 2" should be corrected to "r=200" in Table 6.
2.	In contrast to HETLORA [8], the proposed FlexLoRA needs to perform SVD instead of redistributing it directly before redistributing the aggregated global LoRA weight to clients. This seems to be because it provides better initialization for the low-rank matrices B and A. However, the authors do not mention this point, and this practice has already been presented in FeDeRA [42], so this may not be considered one of this paper's contributions.
3.	Some relevant work in this field is missing, including but not limited to:
[1] Liping Yi, Han Yu, Gang Wang, Xiaoguang Liu, Xiaoxiao Li. FedLoRA: Model-Heterogeneous Personalized Federated Learning with LoRA Tuning. arXiv preprint arXiv:2310.13283, 2023.
[2] Shangchao Su, Bin Li, Xiangyang Xue. FedRA: A Random Allocation Strategy for Federated Tuning to Unleash the Power of Heterogeneous Clients. arXiv preprint arXiv:2310.13283, 2023.

**Questions:**

1.	The method described in Section 3 omits too many preliminaries and algorithmic details, which is reader-unfriendly.

**Limitations:**

1.	The proposed approach essentially fails into the area of federated fine-tuning of LLMs with parameter-efficient fine-tuning (PEFT) techniques. However, this paper does not discuss any limitations that may be imposed by PEFT especially LoRA used in this paper, e.g., PEFT methods sacrifice performance due to the parameter update is limited to a much lower-dimensional subspace, especially when data across clients are non-IID.

---

> ### Author Rebuttal · Authors · 2024-08-07
>
> We appreciate the positive score and valuable feedback from the reviewer! We'd like to respond to the reviewer's questions and comments on the following aspects:
>
> - Paper presentation comments: W1, Question, Limitation
> - SVD contribution: W2
> - Additional related work: W3
>
> ---
> ## Response to Paper Presentation Comments (W1, Q1 & L1)
>
> We deeply thank the reviewer's time for reading our paper and providing suggestions on typos and paper presentation. We will fix the typo issues that the reviewer mentioned in our next version. For section 3, we present our FlexLoRA algorithm pseudo-code and theorem 1 proof in the Appendix B and C sections due to page limit. We will consider moving the algorithm pseudo-code and any relevant details from the Appendix to the main content to make the paper more reader-friendly if our paper is selected and allowed for one additional page. We'd also like to include more discussions regarding the potential performance sacrifice of PEFT methods, as shown in relevant works such as [1].
>
> ## Response to W2 (SVD contribution)
>
> > SVD seems to be because it provides better initialization for the low-rank matrices B and A. However, the authors do not mention this point, and this practice has already been presented in FeDeRA
>
> - [**Algorithm Difference**] We'd like to clarify that **our method differs from FeDeRA**. FeDeRA's approach involves the following steps:
>
> 1. `SVD Decomposition`: Decomposing the model's pretrained weight, $W_0$, using SVD: $SVD(W_0)=U\Sigma V^T$
>
> 2. `LoRA Initialization`: Initializing LoRA with the top r singular values: $B=U[:r, :] \sqrt{\Sigma[:r]}, A=\sqrt{\Sigma[:r]}V^T[:, :r]$
>
> 3. `Freezing Weights`: Using the difference between pretrained weights and LoRA as frozen weights:$\text{freeze} (W_0-BA)$
>
> 4. `Federated Averaging`: Apart from the initialization, the method follows standard federated averaging with LoRA.
>
> This is very different from FlexLoRA, as **FlexLoRA uses SVD for aggregations while FeDeRA uses SVD for their LoRA weights initialization**.
>
> - [**Relationship between SVD and initialization**]: There exists research indicating that the initialization of LoRA can impact its performance (e.g., SLoRA[2]). FeDeRA's improved performance is attributed to better initialization, where the initial LoRA weights contain information from the pre-trained weights, not due to the use of SVD. In contrast, our paper does not claim *SVD as a major contribution*. Our key contribution lies in **enabling clients to scale ranks based on local resources, thereby enhancing performance, and SVD is only utilized for scaling rank to local clients and does not contribute to improving model performance.** In our setting, we have demonstrated that when all clients use the same rank, aggregation with SVD is equivalent to FedAvg, underscoring that the contribution of model performance is not from SVD but an increase in rank.
>
> ---
> ## Response to W3 (Additional Related Works)
>
> > Some relevant work in this field is missing, including: pFedLoRA, FedRA
>
> We thank the reviewer for providing the relevant work.  These works will be cited and discussed in our final version to provide a more comprehensive context. Here is an overview comparison between the work that the reviewer suggests and FlexLoRA:
>
> 1. FedRA: FedRA uses a random allocation strategy for federated tuning to leverage heterogeneous clients. Compared with FedRA which randomly selects several layers for LoRA tuning on each client, FlexLoRA tunes all the layers with LoRA.
>
> 2.  pFedLoRA: pFedLoRA mainly addresses personalization FL with LoRA, which is a different aspect of federated learning. pFedLoRA utilizes both global LoRA and local LoRA weights to boost the local performance of each client, which provides a solution in the personalization area, while our method focuses on an orthogonal optimization perspective: scaling rank adjustment based on local resources. Both approaches have the potential to be combined together to further improve the utility in FL.
>
> Besides, following the reviewer's suggestion to add more relevant works, we newly conduct experiments and would like to provide the result of our supplement experiment on comparing FlexLoRA with FedRA (Table 1 below). From it we can observe that **FlexLoRA is more suitable than FedRA for fine-tuning LLMs in cross-device FL settings**.
>
> |  | Table 12 Result (from paper) | FedRA |
> | --- | --- | --- |
> | Homo Rank | 55.34 | 53.29 |
> | Heavy Tail Light | 58.39 | 56.37 |
>
> Table 1: Result from FlexLoRA vs FedRA
>
> ---
> ## Mentioned Refs
> [1] "Parameter-Efficient Fine-Tuning Methods for Pretrained Language Models: A Critical Review and Assessment." arXiv, 2023
>
> [2] "SLoRA: Federated parameter efficient fine-tuning of language models." arXiv preprint arXiv:2308.06522 (2023).
>
> ---
> ## Closing Remark
>
> We believe the paper has been further improved with your helpful comments, and hope that these responses will convince you to lean more toward acceptance of the paper. We will include the new discussion and results in our final version. Thank you once again!

---

> > ### Comment · Reviewer_ug2H · 2024-08-14
> >
> > I have read the author rebuttal and made any necessary changes to my review.

---

> > > ### Author Response · Authors · 2024-08-14
> > >
> > > Dear reviewer,
> > >
> > > Thank you for your reply and for making the necessary changes! We noticed that the review modification timestamp in the system may be inconsistent. Please let us know if there have been any oversight changes or if you have any further questions. We will do our best to address all your comments. Thank you once again for your time and effort!

---

### Official Review · Reviewer_V4JD · 2024-07-10

**Soundness:** 3
**Presentation:** 2
**Contribution:** 2
**Rating:** 6
**Confidence:** 4

**Summary:**

Under the framework of federated fine-tuning large model, this paper proposes a simple and effective LoRA aggregation method with different ranks, which mainly focuses on the problem of client resource heterogeneity. Specifically, the full-size LoRA is first obtained, and then after normal aggregation, it is decomposed into a matrix BA with the corresponding rank of each client by SVD. Extensive experiments demonstrate the effectiveness of the proposed approach when client resources are inconsistent.

**Strengths:**

1. Different rank-based LoRA is directly matched to client resource inconsistency. The method of first multiplying into full-size LoRA and then decomposing into different ranks based on SVD is simple and effective, and has certain rationality.
2. The information loss of the parameters after SVD decomposition compared with the original global parameters is given in Section 3.4, and this method is generally reasonable.

**Weaknesses:**

1. Does it increase the communication cost to convert matrix BA to full-size LoRA before uploading?
2. It seems that the configurations of ranks in Table 1 are not very diverse. Can the proposed method adapt to very diverse and more flexible configurations of ranks?
3. Different ranks seem to be specific to client resource heterogeneity, but do not seem to be particularly optimized for task heterogeneity?
4. While there are performance comparisons with the very related HETLoRA which is used to solve the resource heterogeneity problem, should comparison results with the rest of the state of the art FL methods also be given.

**Questions:**

As shown in the weaknesses section, we won't add anything more here.

**Limitations:**

This paper lacks a detailed discussion of the limitations and shortcomings of the proposed method.

---

> ### Author Rebuttal · Authors · 2024-08-07
>
> We thank the reviewer's positive score and insightful feedback on our paper! In the following replies, we address all the reviewer's comments point-by-point.
>
> ## Response to W.1
>
> > Does it increase the communication cost to convert matrix BA to full-size LoRA before uploading?
>
> - The current design of our method **does not introduce more communication costs** due to the conversion of matrix BA to full-size LoRA. The local clients directly send their trained B and A matrices to the server, which then handles the full-size computation.
> - **This design is a tradeoff between local and global resource availability.** Typically, in cross-device FL scenarios, local clients are numerous but have limited resources, whereas the server is a device with ample computation power. Therefore, by offloading the full-size computation to the server, we avoid burdening the local clients with the communication cost of transmitting full-size BA matrices.
> ---
> ## Response to W.2
>
> > It seems that the configurations of ranks in Table 1 are not very diverse. Can the proposed method adapt to very diverse and more flexible configurations of ranks?
>
> - **Our method can indeed scale to more diverse and flexible configurations of LoRA ranks.** Our current design utilizes a few typical ranks to validate our method, and the assignment of rank also follows previous works: our baseline FedIT[1] uses rank 8 in their experiment and [2] experiments on both rank 30 and rank 200 in their investigation on LoRA performance.
> - Following the reviewer's suggestion, we conduct supplementary experiments with a more diverse set of rank configurations, similar to those used in HETLORA, to demonstrate FlexLoRA’s adaptability to a broader range of scenarios. We select rank from the following set: $r^i \in \{1,5,10,20,50,100,150,200\} $uniformly (i.e, each rank is selected with the same probability). We conducted experiments on Dolly 15K with the same experiment setting as in section 4.6: by partitioning data into 200 clients with Dirichlet distribution $\alpha=0.5$, and the testing the resulted DataJucier-1.3B model.  From Table 1 below, we can observe that **under a more diverse rank setting, FlexLoRA is still able to maintain its superior performance than regular FedAvg.**
>
> |  | RougeL |
> | --- | --- |
> | Homo Rank | 55.34 (from paper Table 12) |
> | Uniform | 58.56 (from paper Table 12) |
> | Uniform (more diverse rank) | 58.81 |
>
> Table 1: Results of FlexLoRA under a more diverse rank choice.
>
> ---
> ## Response to W.3
> > Different ranks seem to be specific to client resource heterogeneity, but do not seem to be particularly optimized for task heterogeneity?
>
> - We'd like to point out that **using task characteristics to define ranks is not always feasible**. Each client may not have a single task, and determining ranks based on the client’s data distribution could be complex, potentially requiring additional training phases. Although the data partition in our main result in Table 2 is split by task, such a data partition approach is only for mimicking an extreme case of data heterogeneity, and in the real world we anticipate more task overlaps among clients.
>
> - To find a rank suitable for a client's local distribution, one may need to consider prune rank on different weights or layers to optimize its performance [3, 4], which introduces numerous computations. In this case, designing ranks for different tasks is not as meaningful as scaling ranks based on computation resources.
>
> - Our approach, FlexLoRA, was designed for a realistic scenario with both resource and task heterogeneity. **Although ranks are directly scaled by client resources, our method aims to address both resource and task heterogeneity.**
>
> ---
> ## Response to W.4
> > While there are performance comparisons with the very related HETLoRA which is used to solve the resource heterogeneity problem, should comparison results with the rest of the state of the art FL methods also be given.
>
> - While we directly compared FlexLoRA with HETLORA, we also referenced SLoRA(2023) as another method addressing LoRA heterogeneity. We did not compare SLoRA with FlexLoRA as pair-wisely as HETLORA, because *SLoRA is used as one of our FL frameworks that can be incorporated into FlexLoRA*. We'd like to mention one of FlexLoRA’s strengths lies in its **ability to integrate with various existing methods rather than competing directly.**
> - Following the reviewer's suggestion on adding more relevant works, we newly conduct experiments to compare FlexLoRA with FedRA (2024), another SOTA method employs a random allocation strategy to handle resource heterogeneity. In Table 2 below, we can observe that FlexLoRA is more suitable than FedRA for fine-tuning LLMs in cross-device FL settings.
>
> |  | Table 12 Result (from paper) | FedRA |
> | --- | --- | --- |
> | Homo Rank | 55.34  | 53.29 |
> | Heavy Tail Light | 58.39  | 56.37 |
>
> Table 2: Result from FlexLoRA vs FedRA
>
> ---
> ## Mentioned Refs
>
> [1] "Towards building the federatedGPT: Federated instruction tuning." ICASSP 2024-2024 IEEE International Conference on Acoustics, Speech and Signal Processing (ICASSP). IEEE, 2024.
>
> [2] "Towards a unified view of parameter-efficient transfer learning." arXiv preprint arXiv:2110.04366 (2021).
>
> [3] "Efficient personalized federated learning via sparse model-adaptation." International Conference on Machine Learning. PMLR, 2023
>
> [4] "Think locally, act globally: Federated learning with local and global representations." arXiv preprint arXiv:2001.01523 (2020)
>
> ---
> ## Closing Remark
>
> We sincerely hope that our responses have adequately addressed your comments. If you feel they have, we would be grateful if you could consider adjusting the evaluation of our manuscript accordingly. Thanks again for your valuable suggestions! We look forward to any further guidance you may have.

---

> > ### Comment · Reviewer_V4JD · 2024-08-09
> >
> > The author has addressed my concerns, and I agree with raising the score.

---

> > > ### Author Response · Authors · 2024-08-10
> > >
> > > Dear Reviewer,
> > >
> > > Thanks for your quick reply! We are sincerely appreciate for your support in raising the score. Your constructive feedback has been invaluable in enhancing the quality of our work.

---

### Official Review · Reviewer_18q4 · 2024-07-13

**Soundness:** 3
**Presentation:** 3
**Contribution:** 2
**Rating:** 7
**Confidence:** 2

**Summary:**

This paper proposes FlexLora to tackle FL's heterogeneous resources and data distribution problem. FlexLora allows for dynamic adjustment of local Lora ranks by employing SVD for weight distribution, improving the global model’s generalization ability.

**Strengths:**

1.  The experiments are comprehensive. The experiments verified that the proposed method is efficient and improves the generalization.
2.  The paper is well-written and easy to follow.

**Weaknesses:**

1.  This paper claims to follow a cross-device setting, but cross-device setting usually involves thousands of clients, while the client number in the experiment of this paper is much smaller. If the method can only be verified in a small-scale experiment, cross-device setting is not a very rigorous statement. Cross-silo setting might be more suitable for this paper.
2.  One of the main contributions of this paper is allowing for dynamic adjustment of local Lora ranks in FL. Some dynamic/flexible Lora methods already exist in central learning. I wonder whether these methods can also be applied to tackle the heterogeneity of downstream tasks in FL. Related discussions or experiments are suggested here.

**Questions:**

Please refer to weakness.

**Limitations:**

The authors have discussed the limitations of the model and client scale. For the small client scale, claiming a cross-device setting is not so rigorous. Cross-silo setting might be more suitable.

---

> ### Author Rebuttal · Authors · 2024-08-07
>
> We appreciate the reviewer's acknowledge of our contribution! We respond the reviewer's suggestions in the following two replies.
>
> ## Response to W.1
> > This paper claims to follow a cross-device setting, but cross-device setting usually involves thousands of clients, while the client number in the experiment of this paper is much smaller. ... Cross-silo setting might be more suitable for this paper.
>
> - While it is true that our experiments do not reach the scale of millions of clients for cross-device simulation, our study involves over 1,600 clients, which is comparable with existing cross-device FL work. For example, [1] implements cross-device FL benchmarks on the Landmarks dataset with 1,262 clients, and [2] uses more than 700 clients in their cross-device FL setting.
> - Besides, to account for FlexLoRA's scalability on FL settings with larger client pool, we present Figure 5 in our paper, which demonstrates that **the efficacy of our method grows with the client numbers**, showcasing its potential for even larger cross-device federated learning (FL) settings.
>
> ---
> ## Response to W.2
> > One of the main contributions of this paper is allowing for dynamic adjustment of local Lora ranks in FL. Some dynamic/flexible Lora methods already exist in central learning. I wonder whether these methods can also be applied to tackle the heterogeneity of downstream tasks in FL.
>
> - We thank the reviewer for their insightful comment. While dynamic/flexible LoRA methods do exist in centralized learning, their direct applicability to FL settings, especially in handling the heterogeneity of downstream tasks, is not guaranteed.
> - Per your suggestion, we newly conducted preliminary experiments by adapting concepts from one of the centralized dynamic LoRA works, ReLoRA, [3] to federated settings. ReLoRA trains LoRA for several epochs, uploads the LoRA weight to the pretrained weights, then initializes a new LoRA module and trains based on the new frozen weights, repeating this process for several LoRA modules. We create a baseline by incorporating the concept of ReLoRA, uploading aggregated LoRA weights to the pretrained models after several communication rounds.
> - However, we observed a slower convergence speed than the regular FlexLoRA. We show in Table 1 below that incorporating the step of "uploading LoRA weights" does not lead to better performance. This suggests that **observations and improvements seen in centralized dynamic LoRA methods may not translate directly to federated learning due to the unique challenges posed by heterogeneity in FL**, which underlines the importance of tailored solutions like FlexLoRA for effectively managing heterogeneity in FL environments.
>
>
> |  | Homo Rank (Baseline) | Uniform |
> | --- | --- | --- |
> | Table 2 Result (from paper) | 56.53(from paper) | 58.07(from paper) |
> | +ReLoRA Result | 53.9 | 56.56 |
>
> Table 1: Results with/without incorporating ReLoRA into either regular FedAvg or FlexLoRA.
>
> ---
> ## Mentioned Refs
>
> [1] "Motley: Benchmarking heterogeneity and personalization in federated learning." arXiv preprint arXiv:2206.09262 (2022).
>
> [2] "Federated full-parameter tuning of billion-sized language models with communication cost under 18 kilobytes." International Conference on Machine Learning. PMLR, 2023.
>
> [3] "Relora: High-rank training through low-rank updates." The Twelfth International Conference on Learning Representations. 2023.
>
> ---
> ## Closing Remark
>
> We hope our responses can address your comments. Again, we would like to thank you for your  support of our work and detailed comments!

---

### Official Review · Reviewer_XizQ · 2024-07-13

**Soundness:** 3
**Presentation:** 3
**Contribution:** 3
**Rating:** 7
**Confidence:** 4

**Summary:**

This paper proposes a LoRA-based federated fine-tuning algorithm called FlexLoRA. FlexLoRA keeps full-size LoRA modules on the server, and decomposes them into heterogeneous rank LoRA modules according to the client task and capacity. The heterogeneous local LoRA modules will be transferred back to full-size and then aggregated on the server. FlexLoRA can make LoRA-based federated fine-tuning more flexible and easier to be implemented.

**Strengths:**

(1) This paper uses SVD to solve the heterogeneous LoRA rank problem in FL, which looks valid and straightforward. The explanation of the proposed method is clear and easy to follow.
(2) The paper provides sufficient scalability study. With slightly increased cost per round, FlexLoRA can convergence much faster than baselines.

**Weaknesses:**

(1)The main concern is the privacy issue of FlexLoRA. Since the server needs to preset all the local LoRA ranks, the server needs to get access to some client information such as the amount of data, computation resources, ..., etc. Will this threaten the privacy of local data?
(2)The experiment shows results of language model around 1B. Why larger models (e.g., llama) and benchmarks (e.g., mmlu, gsm8k) are not shown in the body part. There are some results in the appendix but may need more detailed discussion in the experiment section.

**Questions:**

please see the weaknesses

---

> ### Author Rebuttal · Authors · 2024-08-07
>
> We thank the reviewer for their insightful feedback and their recognition of our work's contribution to addressing heterogeneity problems in FL! In response to the raised comments, we summarize the two main questions that the reviewer pointed out in the weakness section.
>
> ---
>
> ## Response to W.1
> >  Since the server needs to preset all the local LoRA ranks, the server needs to get access to some client information such as the amount of data, computation resources, ..., etc. Will this threaten the privacy of local data?
>
> Thank you for your comments. The information disclosed from the clients to the server is minimal: **Compared with the vanilla FL method, only the preset LoRA ranks are required to be shared from the clients.** We discuss the influence of disclosure of LoRA rank as the following:
>
> - [**Computation resource**] The LoRA rank is primarily correlated with the computation resources of each client. If the information about computation resources is sensitive to the client, the client can choose to employ encryption techniques or rely on trusted third parties for secure communication.
> - [**Amount of data**] In Federated Learning, the amount of data is usually shared with the server for calculating weight aggregation. **FlexLoRA does not spill more information than vanilla FL methods.** If the amount of data becomes sensitive information in some FL training settings, we can encrypt such information with existing approaches like secure aggregation [1].
>
> ---
>
> ## Response to W.2
> > The experiment shows results of language model around 1B. Why larger models (e.g., llama) and benchmarks (e.g., mmlu, gsm8k) are not shown in the body part.
>
> We answer the reviewer's question about the experiment setting in the following two aspects:
>
> 1. **Model Size:**
>    - We selected a 1B base model for the main experiments to accommodate the real-world cross-device FL setting where thousands of devices are present and a majority of clients are edge devices with limited computation power.
>    - Following the reviewer's suggestion on accounting for FlexLoRA's performance under larger models, we present the results of llama 3(8B) on Dolly-15K dataset with 1000 clients a supplement experiment in Table 1. The client sample rate is 0.01 and we use FedIT as the base framework. We can see that FlexLoRA achieves better performance under uniform distribution and heavy tail light distribution compared with regular FedIT, which assigns homogeneous rank to each client's base model.
>
> 2. **Benchmark:**
>    - For the results under Natural Instructions (Table 2 in the paper), we followed the standard of the original Natural Instruction paper [2], which involves using Rouge-L scores and task-specific evaluations for evaluating model performance. [2] has shown that **Rouge-L is positively correlated with the human evaluation result and is a valid indicator of model language processing performance**.
>
> | **Resource Dist** | **RougeL** |
> | --- | --- |
> | Homo Rank (FedIT) | 68.88 |
> | Uniform | 69.49 |
> | Heavy tail light | 69.29 |
>
> Table 1: FlexLoRA's performance on llama-3 (8B) under 1000 clients.
>
> ---
>
> ## Mentioned Refs
>
> [1] "Practical secure aggregation for federated learning on user-held data." Proceedings of the 2017 ACM SIGSAC Conference on Computer and Communications Security. 2017.
>
> [2] "Super-naturalinstructions: Generalization via declarative instructions on 1600+ nlp tasks." arXiv preprint arXiv:2204.07705 (2022).
>
> ---
>
> ## Closing Remark
>
> We truly appreciate your valuable comments, which have contributed to the improvement of our paper. We hope that our responses demonstrate our commitment to addressing your comments and encourage you to favorably consider accepting the paper. We will ensure that the new discussion and results are incorporated into the final version. Thank you once again for your support!

---

> > ### Comment · Reviewer_XizQ · 2024-08-10
> > **Thank for the response**
> >
> > I very appreciate the authors' response. My concerns have been well addressed, and hence I am willing to raise my score.

---

> > > ### Author Response · Authors · 2024-08-12
> > >
> > > Dear Reviewer,
> > >
> > > Thank you very much for your response! We greatly appreciate your support throughout the review process. Your constructive feedback have been instrumental in helping us refine our work.

---

### Decision · Program_Chairs · 2024-09-25

**Decision:**

Accept (poster)

**Comment:**

This study introduces FlexLoRA which is a federated finetune algorithm based on LoRA. It operates by maintaining a full LoRA module on a central server while decomposing it via SVD as heterogeneous rank LoRA which are tailored to specific client tasks and their capabilities. The authors have conducted thorough experiments and provided a detailed theoretical analysis to demonstrate the proposed method's efficacy and scalability.

All reviewers are positive to the paper and agree that the paper makes significant contributions including it is well-motivated and straightforward and provides comprehensive experiments.

The main concerns raised by reviewers including privacy issues of FlexLoRA, its overhead and lack of a few baselines were properly addressed during the discussion phase.

Overall, I agree with the reviewers and recommend its acceptance